# Improved Correlation Filter Tracking with Enhanced Features and Adaptive Kalman Filter

**DOI:** 10.3390/s19071625

**Published:** 2019-04-04

**Authors:** Hao Yang, Yingqing Huang, Zhihong Xie

**Affiliations:** Department of Arms and Control Engineering, Army Academy of Armored Forces, Beijing 100072, China; huangyingqing1105@163.com (Y.H.); xiezhihong@263.net (Z.X.)

**Keywords:** correlation filter, spatial prior, color histogram, occlusion judgement

## Abstract

In the field of visual tracking, discriminative correlation filter (DCF)-based trackers have made remarkable achievements with their high computational efficiency. The crucial challenge that still remains is how to construct qualified samples without boundary effects and redetect occluded targets. In this paper a feature-enhanced discriminative correlation filter (FEDCF) tracker is proposed, which utilizes the color statistical model to strengthen the texture features (like the histograms of oriented gradient of HOG) and uses the spatial-prior function to suppress the boundary effects. Then, improved correlation filters using the enhanced features are built, the optimal functions of which can be effectively solved by Gauss–Seidel iteration. In addition, the average peak-response difference (APRD) is proposed to reflect the degree of target-occlusion according to the target response, and an adaptive Kalman filter is established to support the target redetection. The proposed tracker achieved a success plot performance of 67.8% with 5.1 fps on the standard datasets OTB2013.

## 1. Introduction

Target tracking is mainly applied in video surveillance, intelligent traffic, unmanned vehicles, and precision-guided munitions, etc. [1,2] The main task is to locate the target given only position information (usually an axis-aligned rectangle) in the video stream. So far, although research on target tracking has made great progress, it is still difficult to build a robust tracker that can meet several challenges (like light intensity changes, target deformation and rapid movement, background clutter, etc.) at the same time without prior knowledge of the target. 

The discriminative correlation filter (DCF)-based algorithms [3,4,5,6,7] are among the most efficient tracking algorithms at present [8]. The main idea is to use the correlation operation to reflect the similarity between the candidate (target sample) and the target template (filter), and then get the most similar sample according to the correlation response peak. Moreover, the correlation operation in the time domain can be implemented in the frequency domain with element-wise operation, which makes the calculation speed of the algorithm much faster than that of other typical trackers [9,10,11]. However, the standard DCF algorithm requires inter-correlation operations between image blocks, so periodic convolution is used to make up the input data [3], which brings boundary effects that makes the target samples only extractable from a restricted search area. In order to solve the boundary effects caused by circulant-shift matrices, many methods have been proposed. CACF [12] adds the local context to the ridge regression model. SRDCF [13] introduces a spatial regularization function into the process of correlation filtering training. BACF [14] constructs a binary mask upon the larger search window. Nevertheless, these methods only suppress the features outside the fixed range of the sample, which means the features of the background and target within the range are not treated differently. Therefore, trackers tend to fail when the texture feature of the target are close to those of the surroundings. 

A feature-enhanced discriminative correlation filters algorithm (FEDCF) is proposed, where the color histogram is used [15], which can distinguishes target and background distractors effectively, to emphasize the texture feature differences between targets and distractors, as there is usually a prominent difference between them in a color model. Considering that part of the background and the target may have similar color model, to suppress the potential impact of this a spatial prior function is constructed to impose a spatial constraint on the sample [16], which is crucial to suppress the boundary effects and combine with color histogram on a feature level. The established weights aim to enhance the target features and suppress the background-distractor features at the same time, which helps to train the filter that is capable of more powerful distinguishing ability. 

In addition, DCF-based trackers can’t handle target-occlusion effectively due to their model updating mechanisms, to address which, the average peak-response difference (APRD) which is sensitive to occlusion is proposed to tackle the target-occlusion with model updating and searching area updating. When a target is judged to be occluded according to APRD, the target model won’t update to retain its purity, and the adaptive Kalman filter [17] based on the APRD parameter is proposed to update the target search area, which aims to ensure that after occlusion the target will still be in the central area where the features are enhanced, rather than in the background area where the features are suppressed by the spatial prior function. The main contributions of the paper are as follows:(1)In order to solve the problem of insufficient expression of texture features (like HOG [18]), a spatial-adaptive weight is constructed by the color-histogram and spatial-prior function, which can increase the discrimination of the texture features between target and background-distractors. (2)Based on a correlation filter framework, a feature-enhanced correlation filter tracker (FEDCF) is proposed. The tracker combines the spatial-adaptive weights with the correlation filter on the feature level, which helps to enhance the target features while suppress the background-distractor features, and thus train the powerful discriminative filter. The Gauss-Seidel iteration [13] is applied to effectively solve the objective function. (3)For target redetection under occlusion, the APRD parameter is proposed, which can be used to reflect the degree of target occlusion and decide model updating. An adaptive Kalman filter based on APRD is proposed to update search area. (4)The proposed FEDCF was compared with the current state-of-the-art trackers on some standard datasets (OTB2013 (Appendix A) [19] and OTB2015 (Appendix A) [20]). The proposed algorithm runs at approximate 5 fps and achieves the top rank in term of accuracy and robustness. 


## 2. Related Works

Target tracking aims to estimate the true trajectory based on initial target position information, and the estimation of the target at each frame may have noise [21,22,23,24,25]. The whole task mainly includes short-time tracking and redetection. The proposed tracker is based on a correlation filtering algorithm, which has been paid great attention because of its dense samples and computational efficiency brought by the cyclic structure. In this section, we briefly review the tracking algorithm based on correlation filtering and the classical target redetection tricks. 

### 2.1. Correlation Filter Based Trackers

The discriminative correlation filter tracker was firstly proposed by Bolme et al. [3]. The method takes the affine transformation of the target grayscale feature as the training sample, and trains the correlation filter that correlates the sample features with the ideal Gaussian response. The target in the next frame can be obtained at the response map peak. MOSSE achieves operation speeds of 600 fps. On this basis. Henriques et al. [26] proposed the Kernelized Correlation Filters (CSK) which takes the circulant shift of the region around the target as the training sample and thus mathematically solves the problem of dense sampling. Therefore, the correlation filter can be learnt by solving the least squares in frequency domain with fast Fourier transform (FFT), and achieves real-time computing speed. Based on this classical tracker, many improved methods have been proposed. As for target features [5,8,27,28,29,30,31], Henriques et al. [4] used multi-channel HOG features to replace grayscale features and introduced a kernel function. Danelljan et al. [7] used a color naming feature (CN) which is dimensionality reduced with PCA. For scale estimation [5,6,32,33], Danelljan et al. [5] used a 32-dimensional scale filter to estimate the scale of the target. However, these correlation filtering algorithms are inevitably affected by the boundary effect. Samples constructed by circulant matrix (see Figure 1), especially at the boundary which can’t represent real feature information. The filter learned from this limited circulant shifted image block is more likely to drift in the case of deformation and motion blur. 

To solve the boundary effects, many methods have been proposed [34,35,36,37,38]. Galoogahi et al. [14] used larger a range of images as training samples to enrich the feature information in the search range, and introduced a binary mask to directly crop the training samples at the center where the information is not affected by the boundary. Based on ridge regression equation, Mueller et al. [12] added constraint spatial context, which essentially suppresses the influence of local context on filter training. However, the contexts selected and the confinement degree are invariant. Danelljan et al. [13] added a regularization penalty term to punish the filter coefficients over a large range surrounding the object. However, the penalty function can only suppress the background information according to its distance to the object, which is not a valid method to distinguish the distractors. Considering that the color histogram has been used for target tracking successfully [15], it could be used to recognize the background distractors as there are prominent differences between them and the target in a color model. 

### 2.2. Target Redetection

Occluded targets can pollute the template filter of DCF-based trackers severely, which is the main reason for tracking loss. How to judge the occlusion of the target, update the template pertinently and redetect the target after occlusion, has been what researchers have focused on. 

There have been several methods proposed for long-term tracking [39,40]. Kalal et al. proposed TLD [41], which decouples the tracking and detection mission, and uses two groups of independent samples to support them, respectively. In the detection stage, the sliding sub-windows are screened according to the variance to find the possible missing targets. MDP [42] applies a template matching method to build a target contour model on the basis of the TLD algorithm. When the target is lost, the model stop updating, and when the model can match the target, the model updates to track. LCT [43] performs estimation of translation and scale for the target, respectively. When the response score is lower than the threshold, the target is judged to be occluded, and the online random ferns is applied to re-determine the optimal target location. Bolme et al. [3] used Peak to Sidelobe Ratio (PSR) to define the peak intensity, and thus to judge the occlusion of the target. However, these methods for target occlusion are either time-consuming or not accurate enough. Based on the DCF tracker, a new confidence index (APRD) is proposed to reflect the degree of occlusion, which can effectively prevent models from being polluted by occlusions. In addition, an adaptive Kalman filter with APRD is proposed. Compared with several improved particle filters [22,23] which aim to estimate the complex state space model or dynamic system, and several improved shadowing filters [21,22,23] which try to optimize the model’s deterministic dynamics including velocity and acceleration, the proposed Kalman filter aims not to model the trajectory of the target but to update the searching area when the model is judged not to update, and as what we will discuss in Section 3.4.2, the step would take almost no extra time. The output of the system adaptively depends on the historical observation from the feature-enhanced correlation filter and the historical predictions from the Kalman filter, which is more consistent with the changing occlusions in real life. 

## 3. Proposed Method

In this section, we introduce our FEDCF tracker. First, in Section 3.1, the overall framework of FEDCF is overviewed. Then in Section 3.2, the feature enhancement method based on color histogram is introduced. The detailed formula of FEDCF is proposed in Section 3.3. Last in Section 3.4, the working principle of APRD confidence index and adaptive Kalman filter is demonstrated. 

### 3.1. The FEDCF Framework

The tracking framework of FEDCF is shown in Figure 2. In the training stage, the input features are exacted and the histogram model is updated according to the searching region obtained in the previous frame. Then the histogram score is combined with the spatial-prior to form the target likelihood, which is used as target adaptive weight. The weight will be used to build new variable z rather than assigned to a feature directly. 

In the detection stage, the input features are correlated with the filter learned in the previous frame to obtain the response map, according to which the predicted target can be obtained. The response map will also be used to build APRD confidence and obtain the maximum value fmzx, which are designed for judging occlusion. If the APRD and fmax are less than their threshold value, respectively, the target is judged to be occluded, then the target model will not be updated until the APRD and fmax are greater than their threshold value. During the period, the adaptive Kalman filter based on APRD will update the search area continually, which aims to guarantee that the target can be located at the center of the searching area. Details of the above are given in Section 3.2, Section 3.3 and Section 3.4. 

### 3.2. Feature Enhancement Based on Color Histogram

In order to improve the distinction of the features between target and background-distractor, we need to strengthen the target texture feature. Considering that the color features are not easily affected even in the case of rapid deformation, the confidence map can be obtained through the Bayesian classifier [15] based on the color histogram, which can reflect the possibility that the pixels in the searching area belong to the foreground of the target. 

Given a target-centered patch I (see Figure 3), the foreground area O is the bounding-box of size $L_{O}\times H_{O}$ and the background area B is the searching area of size M×N. *n_x_* represents the bin *n* of the color component allocated to the patch pixels x∈I. The target likelihood at *x* can be obtained by Bayes’ Rule [15]:
(1)$$P\left(x\in O|O,B,n_{x}\right)\approx \frac{P\left(n_{x}|x\in O\right)P\left(x\in O\right)}{\sum_{\Omega \in \left\{O,B\right\}}P\left(n_{x}|x\in \Omega \right)P\left(x\in \Omega \right)}$$


The conditional probability is obtained from the color histogram, i.e., $P\left(n_{x}|x\in O\right)\approx H_{O}^{I}\left(n_{x}\right)/N_{O}$ and $P\left(n_{x}|x\in B\right)\approx H_{B}^{I}\left(n_{x}\right)/N_{B}$, where $H_{O}^{I}\left(n_{x}\right)$, $H_{B}^{I}\left(n_{x}\right)$ denotes the pixel number of histogram H belonging to $n_{x}$ in the target foreground region and the target background region respectively. $N_{O}$, $N_{B}$ respectively denote the number of pixels in foreground region and background region. In addition, the prior probability can be expressed as $P\left(x\in O\right)\approx N_{O}/(N_{O}+N_{B})$. Equation (1) is rewritten as:(2)$$P\left(x\in O|O,B,n_{x}\right)\approx \frac{H_{O}^{I}\left(n_{x}\right)}{H_{O}^{I}\left(n_{x}\right)+H_{B}^{I}\left(n_{x}\right)}$$


Histogram score $w_{h}$ is obtained from Equation (2) i.e., $w_{h}=P\left(x\in O|O,B,n_{x}\right)$, the size of which is M×N. As Figure 3 shows, $w_{h}$ can enhance the feature information which is similar to the target in the searching area. To prevent the potential influence of the enhanced background-distractor, a spatial-prior function $w_{p}$ of size M×N is defined as follow:(3)$$w_{p}\left(i,j\right)=\frac{1}{1+k\cdot {\left[{\left(\frac{i-x_{B}}{L_{O}}\right)}^{2}+{\left(\frac{i-y_{B}}{H_{O}}\right)}^{2}\right]}^{2}}$$
where (i,j) is the coordinate of pixels x∈I, $\left(x_{B},y_{B}\right)$ is the coordinate of the target center. $L_{O}$, $H_{O}$ denotes the length and height of foreground region respectively. K is the rate parameter for suppressing background. As we can see, $w_{p}$ almost degrades the boundary features at the rate of 1/(1+k), while the features at the center are affected little. The final target likelihood for weighting texture features is then derived below.
(4)$$w_{f}=w_{h}\cdot w_{p}$$


Using *Jogging* for instance (see Figure 3), it can be seen that the color histogram can obviously distinguish a target from its surrounding content. Although, the background-interference which is similar to the target is also enhanced, it can be effectively suppressed by the spatial-prior function (parameter k is set as 5). The target likelihood is finally obtained, which can enhance the target features while suppressing the surrounding-distractors. 

### 3.3. Learning Improved Correlation Filter

In correlation filter based algorithm, circulant samples are generated implicitly which cannot be directly weighted. This is different from traditional machine learning methods, such as SVM [44]. Given a searching area of size M×N, the base sample is a one-dimensional vector $x={\left[x_{1},x_{2},\cdots ,x_{M\times N}\right]}^{T}$, and the cyclic shifting of x by one element is $Px={\left[x_{M\times N},x_{1},\cdots x_{M\times N-1}\right]}^{T}$. All the cyclic shifting of the sample constitute the training sample $X=\;[\;P^{1}x,P^{2}x,\cdots ,P^{M\times N}x\;]$, which has d-dimensional feature vector at $x_{i}\left(m,n\right)\in R^{d}$. The regression target $y=\left[y_{1},y_{2},\cdots ,y_{M\times N}\right]\in R^{M\times N}$ is of Gaussian distribution which decays smoothly from one to zero. The optimal equation of the feature-enhanced correlation filter (FEDCF) is as follows:(5)$$\varepsilon \left(f\right)=\sum_{k=1}^{MN}\left(\alpha_{k}{\left\|\sum_{l=1}^{d}(x_{k}^{l}\cdot w_{f})*f^{l}-y_{k}\right\|}^{2}\right)+\lambda \sum_{l=1}^{d}{\left\|f^{l}\right\|}^{2}$$
where $\alpha_{k}$ denotes the importance of each sample on training the filter, which is uniform in the paper. And λ is a regularization coefficient. According to Parseval’s theory, the Equation (5) can be rewritten in the frequency domain as:(6)$$\varepsilon \left(\hat{f}\right)={\left\|\sum_{l=1}^{d}\left(D(\hat{x^{l}})D({\hat{w}}_{f})\hat{f^{l}}\right)-\hat{y}\right\|}^{2}+\lambda {\sum_{l=1}^{d}\left\|\hat{f^{l}}\right\|}^{2}$$
where $D({\hat{w}}_{f})$ denotes the diagonal matrix with the elements of vector ${\hat{w}}_{f}$ in its diagonal. We define $\hat{X}=\left[\hat{x^{1}}⊕\cdots ⊕\hat{x^{d}}\right]$ and $\hat{W}=\left[D\left({\hat{w}}_{f}\right)⊕\cdots ⊕D\left({\hat{w}}_{f}\right)\right]$ of size dMN×dMN, and $\hat{F}={\left[{\hat{f^{1}}}^{T}\cdots {\hat{f^{d}}}^{T}\right]}^{T}$ is the vectorized filter. The Equation (6) is then rewritten in full vector form:(7)$$\varepsilon \left(\hat{f}\right)={\left\|\hat{X}\hat{W}\hat{F}-\hat{y}\right\|}^{2}+\lambda {\left\|\hat{F}\right\|}^{2}$$


Since $\hat{X}\hat{W}$ is not a circulant matrix which cannot obtain a closed solution directly, we define a new variable $\hat{Z}=\hat{W}\cdot \hat{F}$ of size dMN×1 in the frequency domain. The objective function can be transformed as a cost function of z:(8)$$\varepsilon \left(\hat{z}\right)={\left\|\hat{X}\hat{Z}-\hat{y}\right\|}^{2}+\lambda {\left\|{\hat{W}}^{-1}\cdot \hat{Z}\right\|}^{2}$$


To ensure the fast convergence, the sparse unitary matrix B of dMN×dMN dimension is defined, by which the transformed variable is introduced as follows: $B\hat{X}B^{H}=D$, $B\cdot \hat{Z}=\tilde{Z}$, $B{\hat{W}}^{-1}B^{H}=C$, $B\cdot \hat{y}=\tilde{y}$. Where H represents the conjugate transpose of the matrix. The cost function (8) is then converted into full-real function:(9)$$\varepsilon \left(\hat{z}\right)={\left\|D\tilde{Z}-\tilde{y}\right\|}^{2}+\lambda {\left\|C\cdot \tilde{Z}\right\|}^{2}$$


Like the way that spatially regularized correlation filer [13] is solved, $\tilde{Z}$ is obtained by solving $A\cdot \tilde{Z}=b$ with Gauss-Seidel iterative. Where:(10)$$A=D^{T}D+\lambda C^{T}C$$
(11)$$b=D^{T}\tilde{y}$$


A is composed of strict upper triangle L and lower triangle matrices U, i.e., A=L+U. The iterative process for $\tilde{Z}$ is as follows:(12)$$A_{t}=\left(1-\gamma \right)A_{t-1}+\gamma \left(D_{t}^{T}D_{t}+\lambda C_{t}^{T}C_{t}\right)$$
(13)$$b_{t}=\left(1-\gamma \right)b_{t-1}+\gamma D_{t}^{T}{\tilde{y}}_{t}$$
(14)$$L_{t}{\tilde{z}}^{k}=b_{t}-U_{t}{\tilde{z}}^{k-1}$$


K and t denotes the $k_{th}$ iteration and the $t_{th}$ frame respectively. γ>0 denotes the parameter of the learning rate. At the initial frame, ${\tilde{Z}}_{1}^{0}$ is defined as ${\left(D_{1}^{T}D_{1}+\lambda C_{1}^{T}C_{1}\right)}^{-1}D_{1}^{T}{\tilde{y}}_{1}$ according to Equations (10) and (11), then ${\tilde{Z}}_{t}^{0}$ at $t_{th}$ frame is assigned by ${\tilde{Z}}_{t-1}^{N_{GS}}$. And $N_{GS}$ is defined as the maximum number of iterations. 

At the detecting stage, the sample $\bar{v}$ with the highest response value is selected as the target among the sample set $\left\{v_{1,r},v_{2,r},\cdots ,v_{M\times N,r}\right\}$, where the sample $\left\{v_{k,r}|r\in \left\{\left|\frac{1-S}{2}\right|,\cdots ,\left|\frac{S-1}{2}\right|\right\}\right\}$ is converted from $v_{k}$ at different scales r. S is the total number of scales, and the target $\bar{v}$ is obtained as follows:(15)$$\bar{v}=\underset{v_{k,r}}{\text{arg max}}\sum_{l=1}^{d}(v_{k,r}^{l}\cdot w_{f})*f^{l}=\underset{v_{k,r}}{\text{arg max}}{ℱ}^{-1}\left({\hat{v}}_{k,r}\cdot \hat{Z}\right)$$


The whole complexity of the tracking process is $O\left(dSMN\log MN+\left(d+K^{2}\right)dMNN_{GS}\right)$. Here, d denotes the feature dimension, and MN is the size of the searching area. The total number of scales is S, and k denotes the non-zero term of ${\hat{w}}_{f}$. The computational complexity is mainly derived from the scale estimation and Gauss-Seidel iterative process. 

### 3.4. Target Redetection Mechanism

When the occlusion appears in the searching area, the target template would inevitably be corrupted, however most existing correlation filter trackers [4,5,31] do not consider it when updating the template. In general, the maximum response value of the sample can reflect the confidence of the detected target, but can’t judge occlusion effectively. Therefore, we propose a novel criterion called average peak-response difference (APRD), which can reflect the degree of occlusion sensitively, and helps to decide whether to update the template or not. In addition, according to object’s historical position and state of occlusion, an adaptive Kalman filter is constructed based on APRD, which can provide the most likely searching area for the re-detection of the target after occlusion. 

#### 3.4.1. Model Update

The response map of all circulant samples are obtained as follows:(16)$$f\left(x,w,f\right)={ℱ}^{-1}\left(\hat{X}\hat{W}\hat{F}\right)$$


The maximum and minimum response were defined as $f_{\max}$ and $f_{\min}$ respectively. The APRD parameter is defined as:(17)$$APRD=\frac{\left|f_{\max}-f_{\min}\right|}{\sum_{x}\left|f\left(x,w,f\right)-f_{\min}\right|/{\left(MN\right)}^{2}}$$


Equation (17) is composed of a numerator and a denominator. The denominator implies the average difference of the response map, which mean the similarity of whole samples to the template. The numerator implies the maximum difference of the response map, which mean the confidence of the most likely target candidates. 

Using *Jogging* as an example (see Figure 4) the targets in Figure 4a,c are not occluded yet, so the response maps show one sharp peak, which indicates one definite object. Compared with which, the response map in Figure 4b, where the target suffers from occlusion, shows multiple peaks. Although the peak value does not decrease sharply, the average response difference caused by multiple peaks increases remarkably, and thus APRD decreases significantly. 

Figure 5 shows that the APRD and *f*_max_ have consistent variation trends during the period (64th frame to 81th frame) when the target is occluded, so the occlusion can be judged by these two parameters with a certain threshold. The thresholds are defined as:(18)$$\left\{ \begin{array}{l} {\mathrm{Th}}_{1}=\beta_{1}mean(\sum APRD)\\ {\mathrm{Th}}_{2}=\beta_{2}mean(\sum f_{\max}) \end{array}\right.$$


When APRD and *f*_max_ are smaller than their respective historical average values with ratios $\beta_{1},\beta_{2}$, target is judged to be occluded until the parameters are higher than their thresholds. During the period, the template stop updating and the search area is updated by the Kalman filter. After the occlusion is judged to disappear, the FEDCF tracker redetects the target in the newly updated search area. 

#### 3.4.2. Update the Searching Area

A pure target template is the premise of the target re-detection, and updating the search area reasonably is the key for the target to be redetected successfully after occlusion. Considering that the spatial prior function in Section 3.2 is used to suppress the features surrounding the target box, it is necessary to ensure that the target feature will still be strengthened at the center of the search area after the occlusion. During the period of occlusion, the mission of target prediction could be undertaken by a Kalman filter which is good at using historical target states to predict the target location, of which the observed value are defined as the output of the FEDCF tracker. In addition, since the targets are located with different level of deviation because of their various degree of occlusion, an adaptive Kalman filter is constructed with APRD, which adaptively changes the reliability of the observed values for the Kalman filter. The mathematical model of the Kalman filter is as follows:(19)State equation: X(k+1)=A(k+1,k)X(k)+w(k)
(20)Prediction equations: Z(k)=H(k)X(k)+u(k)


Here, X(k),Z(k) denote the vector of state and observation, respectively. The state transition matrix is defined as A(k+1,k), and H(k) denotes the observation matrix. The vectors of system noise and observed noise are defined as, w(k),u(k) respectively. The predictive equations and update equations are as follows:(21)$$\mathrm{Predictive}\;\mathrm{equations}:\;\{\begin{array}{c} X^{\prime }\left(k+1|k\right)=A\left(k+1,k\right)X^{\prime }\left(k|k\right)\\ P\left(k+1|k\right)=A\left(k+1,k\right)P\left(k|k\right)A^{T}\left(k+1,k\right)+Q\left(k\right) \end{array}$$
(22)$$\mathrm{Update}\;\mathrm{equations}:\{\begin{array}{c} K\left(k+1\right)=P\left(k+1|k\right)H^{T}\left(k+1\right){\left[H\left(k+1\right)P\left(k+1|k\right)H^{T}\left(k+1\right)+R\left(k+1\right)\right]}^{-1}\\ X^{\prime }\left(k+1|k+1\right)=X^{\prime }\left(k+1|k\right)+k\left(k+1\right)\left[Z\left(k+1\right)-H\left(k+1\right)X^{\prime }\left(k+1|k\right)\right]\\ P\left(k+1|k+1\right)=\left[I-k\left(k+1\right)H\left(k+1\right)\right]P\left(k+1|k\right) \end{array}$$


Here, P(k) denotes the prior estimated covariance at the k-th frame, and K(k+1) represents the kalman gain. The covariance of the predicted noise and the observed noise at kth frame are defined as, respectively. 

Because Q(k),R(k) imply the credibility of the predicted value and the observed value for the Kalman filter, respectively, which can be expressed by the APRD parameter. The adaptive Q(k),R(k) at frame t is defined as follows:(23)$$\{\begin{array}{c} R\left(t\right)=1/APRD,\;Q\left(t-1\right)=1-1/APRD\;\;APRD\geq Th_{1}\\ R\left(t\right)=\infty ,\;Q\left(t-1\right)=0\;\;\;\;\;\;\;APRD<Th_{1} \end{array}$$


Here, *Th*_1_ is the threshold of APRD for the template to stop updating. Equation (23) shows that, when the target is judged to be occluded, the Kalman filter no longer accept the observed value coming from the FEDCF tracker, but only predict the target according to the historical state of the target. When the occlusion is not sufficient enough to occlude target, the smaller APRD is, the smaller the effect FEDCF has on the Kalman filter. When the target is judged to reappear after occlusion at t frame, we construct the search area and extraction feature patch based on the current prediction of the Kalman filter, and then enter the detection stage at the next frame. Note that, APRD and *f_max_* are obtained at the detection stage which takes no extra time. The Kalman filter simply uses 3 × 3 matrixes to make one-step prediction at each frame, so the time consumed for the redetection stage is almost negligible. The outline of the whole tracking framework is presented in Table 1. 

## 4. Experiments

We validated out FEDCF tracker on the standard datasets OTB2013 [19] and OTB2015 [20] on MATLAB R2014b. The experiments were carried out on a PC with an Intel I5 2.60 GHz CPU and 8 GB RAM. OTB2013 and OTB2015 contain 50 and 100 video sequences, respectively, with 11 challenging attributes, i.e., illumination variation (IV), out-of-plane rotation (OPR), scale variation (SV), occlusion (OCC), deformation (DEF), motion blur (MB), fast motion (FM), in-plane rotation (IPR), out of view (OPR), background clutter (BC), low resolution (LR). 

The trackers are evaluated with one-pass evaluation (OPE), the evaluation criteria are the precision plots and the success plots. The precision score (PR) is calculated as the ratio of the frames where the Euclidean distance between the estimated target center and the ground-truth center is smaller than a certain threshold. The precision plot is plotted over different thresholds and the threshold in our experiment is set to 20 pixels. The success score is calculated as the percentage of the frames in which the overlap ratio is larger than a certain threshold. The overlap ratio is defined as |a∩b|/|a∪b|, where a, b respectively denote the estimated target bounding box and the ground-truth bounding box. The success plot ranks the trackers according the area under the curve (AUC) of it. 

### 4.1. Detailed Parameters

In our proposed FEDCF method, the size of searching area is empirically set to the 2.5 times the target bounding box. The features exacted at the search area are HOG and CN features, where CN features are used to construct a color histogram with 32 bins in each channel. The parameter *k* in the spatial prior function is set to 5. HOG is the basic feature set used to describe the target, the cell of which is set to 4 × 4 with 31 dimensions, which follows the setting of traditional DCF tracker [5]. In the detection stage, we empirically set the learning ratio *γ* to 0.025 in (12), and the regularization coefficient *λ* to 0.8. The number of iterations *N_GS_* is set to 4, and the number of target scales *S* is 7. In the redetecting stage, $\beta_{1},\beta_{2}$ determine the sensitivity of model updating to the background clutter, and when which tend to 0 or 1, the performance of redetection becomes worse. So according to the experimental tests, both $\beta_{1},\beta_{2}$ are set to 0.5 in (18). Considering that the target starts moving slowly in the initial frames, the kalman filter parameters are set as follows:
$$A\left(k+1,k\right)=\left[\begin{array}{ccc} 1 & 1 & 0.5\\ 0 & 1 & 1\\ 0 & 0 & 1 \end{array}\right],H\left(k\right)=\left[\begin{array}{ccc} 1 & 0 & 0 \end{array}\right],P\left(1|1\right)=\left[\begin{array}{ccc} 1 & 0 & 0\\ 0 & 10 & 0\\ 0 & 0 & 1 \end{array}\right]$$


The whole parameters are fixed over all video sequences, with which the proposed FEDCF is implemented on standard laptop with 5 frames/s. 

### 4.2. Analysis of the FEDCF

Our proposed FEDCF can exact powerful sample features based on a correlation filter framework, where the image patch inevitably suffers from boundary effects. Our proposed improved method can be divided into two aspects, which are strengthening the target and suppressing the background-distractors. Compared with SRDCF, which can be viewed as the state-of-the-art tracker for solving boundary effects, FEDCF improves the DCF tracker at the feature level. First, FEDCF uses a color histogram to reflect the differences between targets and surroundings-distractors in the color model. As we can see from Figure 6, the target in *Bolt* is specified by the red bounding box, and the search area contains background distractors. Because not only target but also the part of distractors which are similar to the target in the statistical color model are enhanced, the spatial-prior is introduced to suppress the distractors which are enhanced by mistake, and the boundary effect is also restrained in the same way. 

In order to prove that our proposed improved methods are effective even in the case of target deformation, FEDCF and SRDCF, KCF [4] are compared in the scenes annotated with deformation in OTB2015. The screenshots of the tracking performances are shown as Figure 7. 

From Figure 7 we can see, the target in *Bird 1* changes its shape drastically, and the targets in *Skating_1*,*2* suffer from interference of background-distractors. FEDCF can track the target persistently and accurately, while SRDCF and KCF drift to the background after a period of tracking. Note that the targets and background-distractors are clearly distinguishable in the statistical color model, and the color histogram of the target is very stable, even when target is undergoing deformation. Although SRDCF constrains the filter coefficients over boundary space where the background-distractors are most likely be, when the distractors surrounding the target are changing shape, SRDCF can’t overcome their interference effectively. 

In order to demonstrate the contributions of each proposed modules, we carried out an ablation experiment on OTB2015. Considering that FEDCF is an improved tracker based on DCF, and the feature-enhancement we used is similar to DAT [15] in principle, so we used DCF [4] as the baseline tracker and treated DAT as the compared tracker. The performances of the trackers are shown as Table 2. 

As we can see, using a color histogram to enhance the HOG features can improve the DCF obviously, though the broken circulant structure obviously increases the computational complexity. However, the performance is inferior to that of SRDCF until the prior spatial function is utilized, because it’s not enough to tackle the boundary effects only by feature-enhancement. Besides which, FEDCF also introduces the adaptive Kalman filter, and the extra module takes nearly no more time, but improves the performance of the tracker in terms of success score and precision score significantly, which validates the effectiveness of our proposed redetection mechanism. Among all the trackers, DAT performs worst, which proves that although the color histogram can make texture features more robust in a complex environment, the single color model without texture features is unable to present the target, especially in the case of deformation and illumination variations where the color information is almost distorted. 

### 4.3. The evaluation on OTB

The proposed FEDCF was evaluated with eight state-of-the-art trackers and one color tracker on the OTB2013 and OTB2015 datasets. The compared trackers include DCF-based trackers: KCF [4], DSST [5], SAMF [6], a tracker that combines color and texture features: STAPLE [31], a long-term trackers good at handling occlusion (LCT [43]) trackers effective at solving the boundary effect like BACF [14], SRDCF [13], and CACF [12], as well as a color tracker which only uses the color histogram (DAT). KCF uses a kernel function and multiple channel HOG features to extend DCF, which can be viewed as a baseline of the DCF-based tracker. STAPLE utilizes the advantages of the color histogram and HOG in describing targets from the color model and texture feature level, and combining them on a response map, which performs better than KCF and DAT [15] individually. DSST and SAMF respectively add the estimation of scale into the target tracking framework with different strategies. Based on DCF, LCT uses an online fern classifier to redetect targets. BACF uses a binary mask on the expanded search area, to achieve samples extractions with no boundary effects. CACF introduces a spatial constraint into the filter learning, which suppresses the potential effect of the local contexts. SRDCF uses regularization function to suppress the filter coefficients with Gauss distribution. All the code of compared trackers were obtained from the respective authors’ websites or open data sources. DAT utilizes a color histogram to construct the color model, according to which it matches the target. 

#### 4.3.1. State-of-the-Art Comparison

The proposed FEDCF is evaluated with precision plots and success plots on OTB2013 with the compared trackers, the results are shown as Figure 8. FEDCF runs on average at 5.1 frames/s. This is because that FEDCF introduces adaptive weights constructed by the color histogram on a *M* × *N* searching area, which destroys the circulant matrix structure, and thus requires Gauss-Seidel iteration to solve the optimal equation. However, the contribution of this high time complexity is that FEDCF is capable of distinguishing the target and surrounding-distractors. As a result, FEDCF ranks first in terms of PR (88.9%) and AUC (67.8%) metrics, respectively. Compared with KCF which is the baseline of the DCF-based tracker, our FEDCF outperforms it by 14.9% and 16.4% in terms of PR and AUC scores. STAPLE fuses the color histogram and HOG features on the response level, which makes the target detection robust to deformation, compared with which, the proposed FEDCF utilizes the color histogram on a feature level, where it works as the weights of the texture features, and thus the spatial prior function can effectively constrain the boundary effects. Although the speed of FEDCF is slower than that of STAPLE, FEDCF performs better than STAPLE by 10.7%/8.5% in terms of PR/AUC. SRDCF is famous for suppressing boundary effects, and our FEDCF outperforms it by 5.1%/5.0%. LCT, which is good at solving occlusion problems in long-term tracking, shows inferior performance than FEDCF by 4.1%/8.5%. In addition, the proposed FEDCF performs better than other DCF-based trackers such as SAMF, DSST, CACF and BACF. DAT which only uses color model for target tracking performs worst among all the trackers, and FEDCF surpasses it by 47.8%/36.4%, which demonstrates that the target tracking in a complex environment depends more on texture features than on color features, thus it’s reasonable to utilize the color histogram as the weights for texture features. Overall, the proposed FEDCF achieves highly competitive performance against state-of-the-art trackers on OTB2013. 

For a more reliable evaluation of our FEDCF, we use OTB2015 which expands the number of challenging video sequence to 100. The overall precision plots and success plots of FEDCF and compared trackers with one pass evaluation are shown in Figure 9. The proposed FEDCF still ranks first against other seven state-of-the-art trackers. FEDCF achieves precision plot PR score of 86.4% and success plots AUC score of 65.4%, which outperforms the second tracker SRDCF by 7.6%/5.6%. As for the compared trackers, SRDCF, BACF, STAPLE perform better than others, which means our FEDCF is more robust than them. 

#### 4.3.2. Attribute-Based Evaluation

For a more detailed analysis of trackers’ performances in different scenarios, the one-pass evaluations of trackers in term of precision plots on different challenging scenarios are shown as Table 3. 

Our approach ranks first among all the trackers in ten challenging scenarios. The evaluated scores are respectively illumination variation (83.7%), out-of-plane rotation (85.0%), scale variation (84.0%), occlusion (81.0%), deformation (84.1%), motion blur (82.6%), fast motion (80.2%), in-of-plane rotation (81.1%), out-of-view (76.6%) and background clutter (87.2%). Note that, most contributed low resolution sequences are gray sequences, which do not support building a color model. Thus, both FEDCF and STAPLE, which utilize the color histogram, have worse performances in the low resolution challenge compared with their performances on the other ten challenges. However, among the eight state-of-the-art trackers, FEDCF ranks second on the low resolution challenge. Compared with SRDCF, our FEDCF performs more robustly, especially when undergoing occlusion, deformation, and motion blur, which proves that our proposed methods can significantly improve the performance of DCF-based trackers. This is attributed to two reasons. One of the reasons is that the adaptive weights constructed from the color histogram and spatial prior could effectively enhance the target features even when undergoing deformation, and another reason is our proposed method of dealing with occlusion, which show that using the predicted target location from the adaptive Kalman filter to update the search area during occlusion is feasible, and can be applied to other DCF-based trackers which have small search area sizes. Overall, our proposed method shows more competitive performance compared with other DCF-based trackers. 

In order to intuitively show the compared performances of the trackers on challenging video sequences in the OTB2015 dataset, we show screenshots of eight challenging videos from OTB2015 (see Figure 10). Because of space limitations, we only compare our FEDCF with the top four trackers: BACF, SRDCF and Staple, and their estimated bounding boxes are respectively shown as the red, green, black and blue rectangular boxes. The sequences in first two row (*Blurcar1, BlurOwl*) mainly suffer from serious motion blur. In the *Blurcar1* image, BACF and STAPLE lost the target at the beginning and drift to the background, and only our FEDCF and SRDCF finish the tracking mission with great ability by utilizing HOG features. The target in *BlurOwl* undergoes even more serious motion blur where the target feature is kind of similar to the background, and BACF still loses the target at the start. As we can see, STAPLE hardly catches the target by matching with the color histogram. As for SRDCF, which only relies on texture features, the estimation of scale is not accurate enough, however, our FEDCF can keep up with the target accurately. The sequences of the next four rows (*Box, Human3, Jogging, Lemming*) can mainly be considered occlusion scenes. When the target in *Jogging* is occluded, only our FEDCF and SRDCF redetect the target successfully after its reappearance. The success of SRDCF can be attributed to its larger searching area which is four times the target size and that the moving distance of the target is not too long in the plane because of the moving camera. Although the search area of our FEDCF is smaller (2.5 times the target area), it still performs well. In the *Human3* scene where the camera is fixed, not only BACF and STAPLE, but also SRDCF lost target the finally. Our proposed FEDCF is the only tracker which doesn’t lose the target. This is because FEDCF utilizes the Kalman filter to update the search area which ensures that the target will still be located at the center of the search area where the features are enhanced after occlusion. The target in *Lemming* undergoes a brief occlusion, which only FEDCF and BACF can tackle well. The scene of *Box* is similar to *Lemming*, however the target suffers from occlusion for over 10 frames. Only FEDCF catches the target successfully all the way, which is due to the fact that FEDCF uses APRD to stop updating the template immediately when the target is occluded and effectively prevents the template from being corrupted. The sequences of the last two rows (*Shaking, Skating _1*) mainly suffer from illumination variation and deformation. In the *Shaking* scene, SRDCF and STALE lost the target after drastic lighting changes, while our FEDCF and BACF show great resistance. As to *Skating _1*, BACF and STAPLE lost its target which undergoes deformation, then SRDCF drifts to the background due to the illumination variation. Only our FEDCF keeps its tracking box around the target, which proves that under challenging scenarios like illumination and deformation, the texture features like HOG are insufficient to express the target. However, colors histogram and spatial priors can effectively enhance the difference between the target and background-distractors, which is hardly affected by deformation and illumination variation. In conclusion, our proposed methods of adaptive weights and redetection methods can successfully handle different challenges, especially occlusion, deformation, and motion blur. 

## 5. Discussion and Conclusions

A feature-enhanced discriminative filter (FEDCF) is proposed for visual tracking. A novel method using color histograms and spatial prior functions to construct the adaptive features is designed to enhance the differences between the target and background distractors on a feature level. In addition, an effective target redetection mechanism is proposed, which uses the proposed APRD parameter to judge model updating and construct an adaptive Kalman filter to effectively handle target occlusion. Comparative experiments demonstrate that FEDCF can handle the most challenging scenes in OTB2013 and OTB2015. Compared with SRDCF, STAPLE and BACF, the proposed approach performs better, especially under the scenarios of deformation, motion blur and occlusion. Moreover, the proposed method for target occlusion can be applied to other DCF-based trackers to improve their robustness in complex environments. Our future work will focus on simplifying the optimal function and improving the speed of the algorithm. 

## Figures and Tables

**Figure 1 sensors-19-01625-f001:**
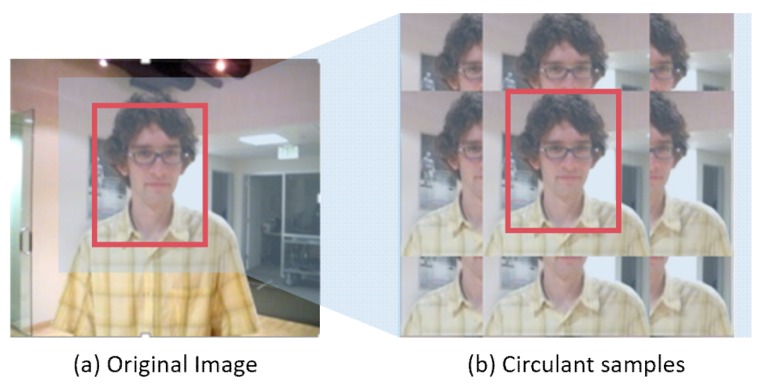
Image (**b**) shows the circulant samples obtained by DCF methods from original image (**a**), leading to a limited range of features for the filter to be learnt from. As a consequence, the template learnt is not discriminative enough.

**Figure 2 sensors-19-01625-f002:**
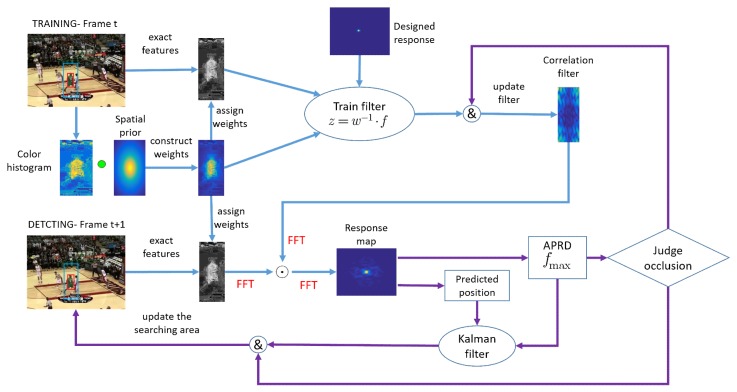
Main components of our tracking framework. The whole improved methods are represented by blue lines and purple lines. The blue lines denote the improving correlation filter with adaptive weights, which aims to enhance the target feature and suppress the background distractors. The purple lines denote the treatment to occlusion, which can prevent the target template from occlusion pollution and ensure that the target after occlusion can be redetected at the updated searching area.

**Figure 3 sensors-19-01625-f003:**
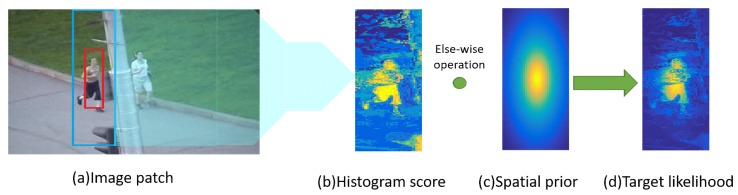
The input patch is shown as (**a**), where the foreground and background of the target are extracted from the red and blue bounding boxes, respectively. (**b**) shows histogram score which is obtained according to the color model in the search area and can reflect the differences between the target and background interference. Spatial prior at (**c**) indicates that the features at the center are more important than the features around. Combining the histogram score and spatial prior, the target likelihood can enhance the target features while suppressing the surrounding-distractors, which is shown as (**d**).

**Figure 4 sensors-19-01625-f004:**
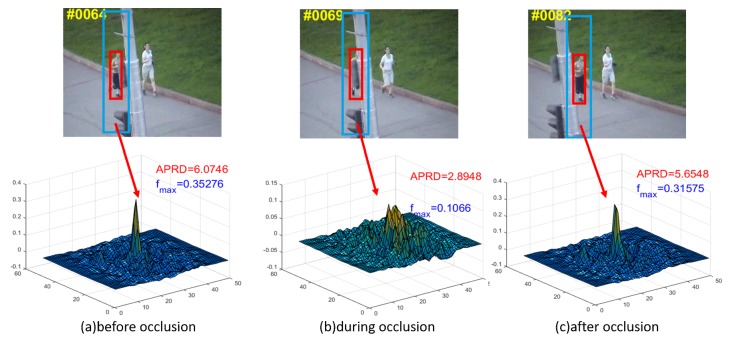
The first row are the shots of the sequence *Jogging* from OTB2013, where the red and blue bounding box denote the target and the searching area of FEDCF, respectively. The second row are response maps corresponding to the scenario above. (**a**), (**b**) and (**c**) respectively show the scenarios before occlusion, during occlusion and after occlusion.

**Figure 5 sensors-19-01625-f005:**
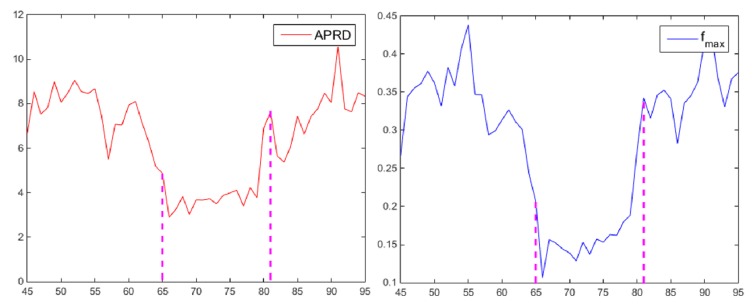
The blue curve and the red curve above respectively show how *f*_max_ and APRD varies between the 45th frame to the 95th frame. The occlusion occurred between the 64th frame to the 81th frame.

**Figure 6 sensors-19-01625-f006:**
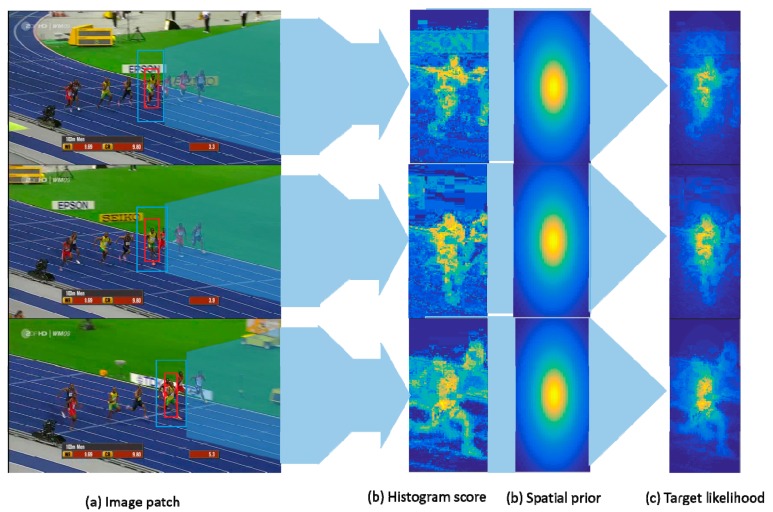
Column (**a**) displays the screenshots from *Bolt* at frames 75, 90 and 125. The target and search area are respectively specified by the red and blue bounding boxes. The effects of enhancements with histogram scores are visualized as column (**b**). The final weights assigned to features are shown are column (**c**).

**Figure 7 sensors-19-01625-f007:**
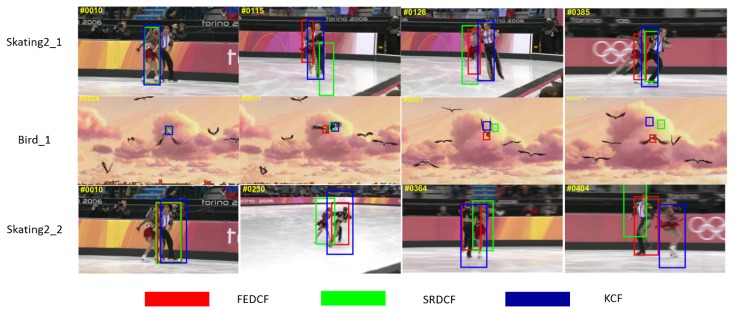
The screenshots from *Skating_1*, *Bird 1* and *Skating_2*, where the targets are the female skater, bird at the screen center and the male skater, respectively. The target results of FEDCF, SRDCF and KCF are respectively highlighted with red, green and blue bounding boxes.

**Figure 8 sensors-19-01625-f008:**
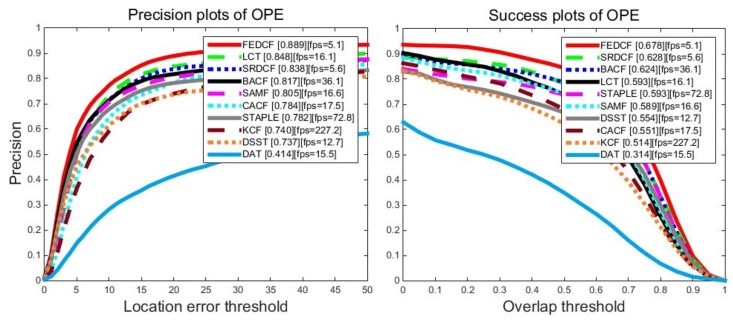
The precision plots and success plots of FEDCF and compared trackers with one-pass (OPE) evaluation on OTB2013, where 50 video sequences contain 11 challenging attributes.

**Figure 9 sensors-19-01625-f009:**
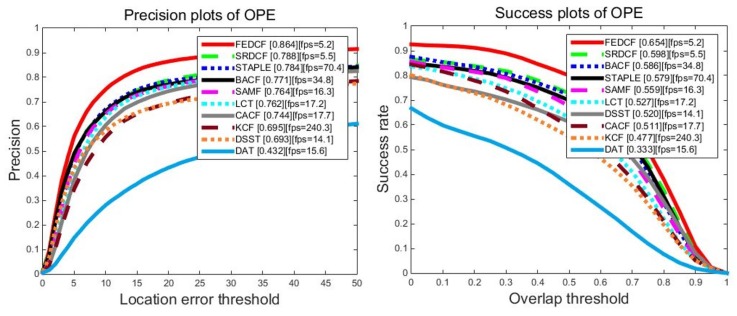
The precision plots and success plots of FEDCF and compared trackers with one-pass (OPE) evaluation on OTB2015, which expands the contained sequences to 100 challenging sequences with 11 challenging attributes.

**Figure 10 sensors-19-01625-f010:**
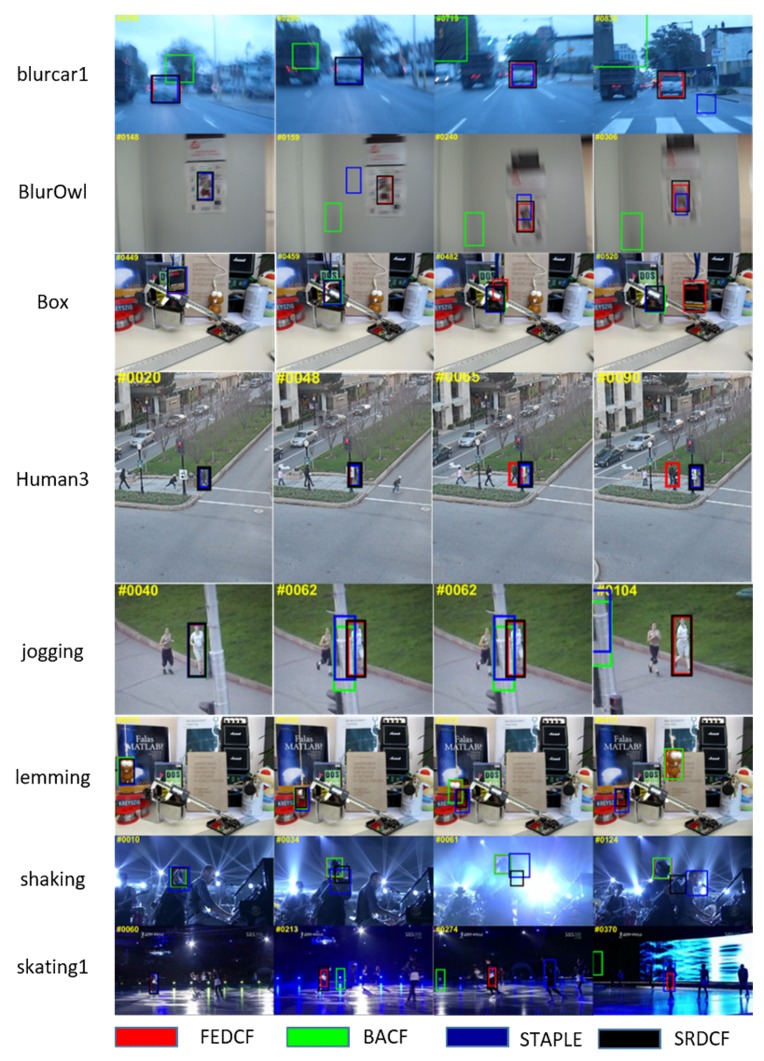
The screenshots of the tracking process on eight challenging sequences. The results of trackers (FEDCF, BACF, STAPLE and SRDCF) are respectively marked with red, green, blue and **black** rectangles.

**Table 1 sensors-19-01625-t001:** The general flow of the whole FEDCF tracking algorithm.

**Input:**	Frames ${\left\{I_{t}\right\}}_{1}^{T}$, initial target location *x*_1_, spatial prior *w_p_*.
**Output:**	Detected target location at each frame ${\left\{x_{t}\right\}}^{T}$.
1:	**repeat**
2:	Crop a searching patch *S_t_* which is centered at *x_t_*.
3:	Exact feature samples V*_t_* and construct the color histogram model via (2).
4:	Construct the adaptive weights *w_t_* via (4).
5:	Obtain APRD and *w_f_* according to (15)–(17).
6:	**If** target is judged to be occluded via (18).
7:	Predict the target location *x_t_*_+1_ by adaptive Kalman filter via (19)–(23).
8:	**Else**
9:	Update the correlation model via (12)–(13).
10:	Detect the target location with certain scale *x_t_*_+1,*r*_ via (15).
11:	**End if**
12:	**Until** end of video sequence.

**Table 2 sensors-19-01625-t002:** The evaluation of OPE in term of scores of success and precision plots on OTB2015. The top score is marked in red.

Trackers	Success Score (%)	Precision Score (%)	Frames Per Second
DCF	47.7	69.5	240.3
DCF+ enhancement	55.9	76.2	14.9
DCF+ enhancement+ prior spatial	62.7	81.6	5.3
FEDCF	65.4	86.4	5.2
SRDCF	59.8	78.8	5.5
DAT	33.3	43.2	15.6

**Table 3 sensors-19-01625-t003:** The one-pass evaluation of FEDCF and compared trackers in term of PR score on OTB2015. The top three scores are highlighted in red, green and blue respectively.

	FEDCF	SRDCF	STAPLE	BACF	CACF	DSST	SAMF	KCF	LCT	DAT
Iv(37)	83.7	78.1	77.8	75.8	76.0	72.0	74.0	72.4	74.3	33.4
OPR(63)	85.0	74.0	73.8	75.3	69.2	66.5	75.0	67.6	74.6	45.4
SV(63)	84.0	74.3	72.4	75.6	69.4	65.3	72.2	63.5	67.8	41.4
OCC(48)	81.0	72.7	72.4	69.6	66.9	60.2	73.6	63.2	67.8	42.4
DEF(43)	84.1	73.0	74.7	72.6	69.2	56.0	67.1	61.9	68.5	46.9
MB(29)	82.6	76.7	69.9	58.4	71.1	57.0	68.7	60.0	66.9	34.3
FM(39)	80.2	76.9	71.0	68.9	72.7	57.5	70.6	62.1	68.1	38.1
IPR(51)	81.1	74.2	76.8	76.0	75.1	71.1	74.4	70.1	78.1	44.3
OV(14)	76.6	60.2	66.8	74.8	57.8	48.0	67.6	49.9	59.2	36.5
BC(31)	87.2	77.5	74.9	80.1	76.4	70.4	72.2	71.2	17.2	37.1
LR(9)	73.7	66.3	61.0	74.1	59.4	60.2	68.4	56.0	53.7	39.3
Overall	86.4	78.8	78.4	77.1	74.4	69.3	76.4	69.5	76.2	43.2

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
