# Peer review of "Improved Correlation Filter Tracking with Enhanced Features and Adaptive Kalman Filter"

_sensors, 2019, doi:10.3390/s19071625_

Round 1

Reviewer 1 Report

Summary
The paper presents an alternative tracker named feature-enhanced discriminative correlation filter (FEDCF) for visual object tracking. The proposed tracker is specifically designed to eliminate the boundary effect and recover the target from the tracking failures caused by occlusions. The work is interesting, the paper is well written and easy to follow, and the results are convincing to reflect the  contributions in this work.

Novelty
The contribution of this paper is mainly in the following two aspects. First, A novel criterion named
APRD is proposed to determine whether occlusion occurs. Second, an adaptive kalman filter is adopted to recover the target from tracking failures caused by occlusions.

Need Improvement

Few minor issues should be addressed. For example:

(1) Abstract - "The proposed tracker achieved the optimal performance (84.5%AUC) with 5.1fps on standard datasets OTB2013." To the best of my knowledge, the performance of FEDCF tracker is not currently the optimal (best) one. Therefore, I suggest the authors to rephase this sentence in Abstract and also cite few state-of-the-art works in the main body of this paper.

(2) I suggest the authors to add a few sentences or tracked frames to discuss, where the method  does not work so well or could be failed. In this case, it is better to know how this work can further be improved.

(3) There are some grammatical errors and need to check on the vocabulary.

Author Response

.

Reviewer 2 Report

Reviewing article “Improved correlation filter tracking with enhanced features and adaptive Kalman filter

Sensors-456967

Decision: The authors propose an improvement for discriminative correlation filter (DCF) based trackers to enhance the visual tracking. They propose to use colour histogram and spatial prior function to better distinguish between the target and background distractors. They also propose adaptive Kalman filter to treat target occlusion effect and redetect the target. The proposed procedure sounds great and evidently enhances the tracking purposes. However, there are major problems with the paper as currently presented and I will not recommend it to be published in Sensors until some improvements are made to the presentation of the paper. Please find below the comments.

Major comments:

There are a lot of abbreviations have been used without clear definitions or references. Wherever you first use an abbreviation, either define or give a relevant reference.

There are a lot of English grammar mistakes (some are listed below).

The abstract is preferable to be in plain language; need to define HOG feature in the abstract; the last sentence in the abstract (lines 19 & 20) is very detailed for abstract (enough to state that the proposed tracker outperforms the-state-of-the-art trackers.

Line 29, give a reference to support your claim.

Paragraph between lines 45 to 54 needs to be written in better way; the sentences are too long and confusing (some grammatical mistakes). Also use references when you first propose colour histogram and spatial prior function.

Line 59, give a reference for adoptive Kalman filter. Line 72, give a reference for Gauss-Seidel iteration.

Line 80, very specified sentence, target tracking might have many tasks and

pusposes. (please see the references I provide below in point 8 for tracking using different filters)

Subsection 2.2: literature review for target redetection and filters. There are other filters rather than Kalman filter are used successfully for long-term target tracking and redetection, such as,

Particle filter: [1]” Gustafsson, F.; Gunnarsson, F.; Bergman, N.; Forssell, U.; Jansson, J.; Karlsson, R.; Nordlund, P. Particle Filters for Positioning, Navigation and Tracking; Linköping University Electronic Press: Linköping, Sweden, 2001.” 

[2] “Míguez, J. Analysis of Selection Methods for Cost-reference Particle Filtering with Applications to Maneuvering Target Tracking and Dynamic Optimization. Digit. Signal Process. 200717, 787–807”

And Shadowing filter:

[3]“Zaitouny A, Stemler T., Algar S. Optimal Shadowing Filter for a Positiong and Tracking Methodology with Limited Information. Sensors2019, 19”

[4] “Zaitouny, A.A.; Stemler, T.; Judd, K. Tracking Rigid Bodies Using Only Position Data: A Shadowing Filter Approach Based on Newtonian Dynamics. Digit. Signal Process. 201767, 81–90.”

[5]“Zaitouny, A.A.; Stemler, T.; Small, M. Tracking a Single Pigeon Using a Shadowing Filter Algorithm. Ecol. Evolut. 20177, 4419–4431. “

Enrich your literature review by considering these references, and provide a brief discussion why you proposed adoptive Kalman filter for this task rather than particle or shadowing filters for example.

Line 151, define f_max

Equation 1: use brackets, it is confusing as presented.

Line 174, use brackets for the equations

Line 205 and 206: what is IR? I believe it is the real number set. If so, please use the correct symbol.

Equations 5 and 6, use brackets for the sum symbol.

Line 224,” .... and lower triangle U matrices”

Section 4: please give a brief reasoning why you chose the thresholds and parameters as stated.

Line 364: Please explain with more details how STAPLE tracker is different than your proposed tracker, as STAPLE uses colour histogram as well but with HOG instead of prior function, how does this enhance the tracking?

From line 410 to 414: [Note that, ..... features from]. This was not very clear, please give more explanation.

Minor issues: (some not all)

Lines 27 and 28, put the examples of the challenges between brackets.

Line 60, “...which is aim...” replace with “... which aims...”

Line 112, add comma after however.

Line 133: “the model is judged”

Line 155: “details of above”

Line 185: “ degrades”

Line 196: “indicates that the features at centre is more important ...”

Line 239: “appears”

Lines 255, 256,257: “Compared with .... peaks” the sentence does not sound

grammatically correct nor clear, please edit.

Line 266: “show” no ‘s’

Line 268: “Fig.5” no “the”

Line 272: “judged” and “thresholds”

Line 284: “targets”

There are some other similar stuff, please give a critical read and try to fix them

Author Response

.

Reviewer 3 Report

The study presented a tracking algorithm which switches between two trackers based on the occlusion detection signal. The first tracker is a "color-based model-free" tracker (from [33]) which is enhanced using spatial regularization of SRDCF, and the second tracker is an adaptive Kalman filter which continues tracking the invisible target during occlusion. The occlusion signal is novel and well-justified and it seems to be working well for this task.

The flow of the paper, literature review and the technical details are good. The experiment section is appropriate, yet still could be improved. The manuscript has a few typos within text and figures  (eg. Fig2 DETCTING -> DETECTING)

* An ablation study experiment can be conducted to clarify the contributions of each module in the overall performance of the tracker:

- baseline tracker [33]

- baseline + spatial reg

- baseline + spatial reg + update stop (when occluded)

-  full method (baseline + spatial reg + update stop + KF tracker)

* Beta 1 and 2 should be discussed and their tuning approach should be explained since they include both threshold and normalization factor.

* Table 2 show include a row for baseline tracker ([33]). Also, Table 3 should have this tracker as a column and the improvement over that should be discussed.

Author Response

.

Round 2

Reviewer 2 Report

Reviewing article “Improved correlation filter tracking with enhanced features and adaptive Kalman filter

Sensors-456967

Decision: The authors propose an improvement for discriminative correlation filter (DCF) based trackers to enhance the visual tracking. They propose to use colour histogram and spatial prior function to better distinguish between the target and background distractors. They also propose adaptive Kalman filter to treat target occlusion effect and redetect the target. The proposed procedure sounds great and evidently enhances the tracking purposes. 

The revised version significantly enhanced the presentation of the manuscript. Consequently, I accept this paper to be published in Sensors.